# Nurse Practitioner Care Delivery Models: Meeting the Rapidly Expanding Needs of Cancer Patients

**DOI:** 10.3390/curroncol32090492

**Published:** 2025-09-02

**Authors:** Tammy O’Rourke, Marcie Smigorowsky, Danielle Moch, Tara Hoffman, Krista Rawson, Teresa Ruston, Julia Beranek, Cindy Railton, Cecilia Joy Kennett, Calvin P. Kruger, Shuang Lu, Nanette Cox-Kennett, Edith Pituskin

**Affiliations:** 1Faculty of Health Disciplines, Athabasca University, Athabasca, AB T9S 3A3, Canada; tammyorourke@athabascau.ca; 2College of Registered Nurses of Alberta, Edmonton, AB T5S 1P2, Canada; msmigorowsky@nurses.ab.ca; 3Faculty of Graduate Studies and Research, University of Alberta, Edmonton, AB T6G 2E1, Canada; dlperrea@ualberta.ca; 4Alberta Health Services, Calgary, AB T2N 5G2, Canada; tara.hoffman@albertahealthservices.ca (T.H.); krista.rawson@albertahealthservices.ca (K.R.); teresa.ruston@albertahealthservices.ca (T.R.); julia.beranek@albertahealthservices.ca (J.B.); cindy.railton@albertahealthservices.ca (C.R.); shuang.lu@albertahealthservices.ca (S.L.); nanette.coxkennett@albertahealthservices.ca (N.C.-K.); 5Faculty of Nursing, University of Alberta, Edmonton, AB T6G 1C9, Canadacpkruger@ualberta.ca (C.P.K.)

**Keywords:** nurse practitioner, chemotherapy, administration, advanced practice, survivorship, targeted therapy, adjuvant breast cancer, autologous bone marrow transplant, radiation oncology, medical oncology, chronic lymphocytic leukemia

## Abstract

In the last decade, highly effective and complex cancer therapies are available to successfully treat the rapidly increasing numbers of cancer cases. Therefore, the need for patients to receive both an early diagnosis and ongoing care has never been so important. Nurse practitioners possess a minimum of a master’s degree and extensive oncology experience. The purpose of this article is to describe four successful nurse practitioner care delivery models: the Assigned model, Consultative model, Partner model, and Most Responsible Provider, significantly contributing to enhanced and expanded cancer care delivery. Taken together, oncology nurse practitioners hold great potential to address the increasingly complex care needs of a rapidly growing population of cancer patients and survivors, today and in the future.

## 1. Introduction

Half of all Canadians will develop cancer at some point in their lifetimes [1]. These rates have increased substantially over the last decade alongside increasing effectiveness and complexity of treatment options. Therefore, the need for patients to receive both an early diagnosis and ongoing care has never been so important. In Alberta, referrals to oncology have increased 18% in the last 7 years with no commensurate increase in the number of oncology specialists [2]. Accordingly, the number of Alberta cancer patients receiving initial consultation outside the recommended time by the Provincial Cancer Guidelines has escalated by 68% [3]. Even a four-week delay of treatment initiation, regardless of modality (surgical, systemic, or radiotherapy) is associated with increased mortality in many cancers [4]. Challenges with oncologic care access and provider recruitment are not unique to Alberta. Many provinces report difficulties in the recruitment and retention of specialty medical (MO) or radiation (RO) oncologists. Factors include, but are not limited to, geography, lifestyle, financial, professional, and academic opportunities [5,6]. A potential solution to such issues is to integrate another type of health care provider. Nurse practitioners possess a minimum of a Masters’ degree alongside extensive oncology experience. In 2004, Cancer Care Alberta, specifically the Cross Cancer Institute (CCI), embarked on an initiative focusing on nurse practitioner (NP) care provision aiming to address these gaps. The purpose of this article is to describe four NP care models: the Assigned model, Consultative model, Partner model, and Most Responsible Provider (MRP model), significantly contributing to enhanced and expanded cancer care delivery at CCI. We aimed to describe the contributions of oncology specialist NPs in various cancer types and care pathways.

## 2. Methods

*Assigned model:* This model was the first to be adopted (2004) at CCI in early breast cancer (EBC) adjuvant treatment. At the initial oncology consultation, the multidisciplinary team performs a broad review of the case and determines systemic therapy recommendations and potential clinical trials. During the in-person consultation, the MO advises the patient that EBC care is provided by a team, and the NP will be their assigned care provider from this point forward. The patient is subsequently scheduled in the Assigned NP clinic, where all systemic therapy, hormone therapy, diagnostic imaging and labs, necessary consultations, and any referrals to supportive care are determined and ordered by the NP. The MO is contacted if an issue should arise requiring team decision-making.

Previously, we undertook a review of the Assigned model in comparison with two others (Available and Rural) in Alberta. With the Available model at the southern tertiary center, patients had access to a mixed-provider group (MO, NP, or Physician Associate). The Rural model represented a variety of providers (general practice physician or MO) in community cancer centers. Women diagnosed with stage I and II breast cancer, between 1 January 2012 and 31 December 2013, receiving systemic therapy and surgery (either order) were identified from the Alberta Cancer Registry using the International Classification of Diseases for Oncology 3rd Edition coding. Given the high cure rates of EBC [7], the outcomes of interest were average chemotherapy dose intensity and chemotherapy completion. Data were collected from the Cancer Care EMR, with dose intensity and treatment completion calculated based on the projected versus actual dose and cycle administered for each chemotherapy agent.

*Dose Intensity = (Actual dose received* ∗ *Actual cycle received)/(Highest dose received* ∗ *Planned cycle)*


*Treatment Completion = Actual cycle received/Planned cycle*


The comorbidity score was estimated and assigned based on aggregated clinical risk scores (ACRGs) obtained from hospital admissions. The ACRGs not only categorize individuals’ illnesses but also address the severity of illness and have been validated against the Charlson index [8,9]. The ACRGs were collapsed into 4 categories of increasing comorbidity: 10–19 = 1, 20–49 = 2, 50–69 = 3, and 70–99 = 4 for risk outcome analysis [10]. Multivariate logistic regression for the associated odds of receiving high- (≥90%) [11,12] or low-dose intensity were assessed by the three models of care, adjusted for the various factors such as patients’ age at diagnosis, clinical risk group prior to diagnosis, cancer stage, ER, PR, HER2, as well as lymphovascular invasion. A similar analysis was also performed for the associated odds of receiving complete treatment (100%) or not. The model was tested for goodness-of-fit using the Hosmer and Lemeshow test [13]. SAS statistical software version 9.1 [14] was used for data management and analyses. A *p*-value < 0.05 was considered for all statistical significance. Ethical approval was secured from the University of Alberta Research Ethics Office, file Pro00055590.

Descriptive statistics are shown in Table 1. The majority (47%) of EBC patients attended the Assigned Model at CCI. Clinical characteristics and patient complexity were not significantly different between any model. Clinical and safety outcomes are shown in Table 2. Average chemotherapy dose intensity and completion were equivalent in the Assigned and Available models. A dose intensity of <90% was observed and was statistically significant in the Rural model, possibly due to the variability of health care providers available. After adjusting for demographic and clinical factors, e.g., patients’ age at diagnosis, clinical risk group prior to diagnosis, cancer stage, ER, PR, HER2, as well as lymphovascular invasion, the odds of receiving high- (≥90%) intensity treatments in the Available model was higher than the Assigned model (OR: 1.3, CIs: 1.0–1.8), whereas the odds of receiving high-intensity treatment in the Rural model was slightly lower than the Assigned model (OR: 0.7, CIs: 0.5–1.1). After adjusting for the factors mentioned, the odds of treatment completion remained equivalent in the Assigned and Available models (OR: 1.0, CIs: 0.7–1.4), but lower in the Rural area (OR: 0.7, CIs: 0.4–1.0).

To the best of our knowledge this is the first report of NP-led care in medical oncology demonstrated as safe and equivalent to that of MO providers specific to patient complexity, chemotherapy dose intensity, and treatment completion. These data convincingly demonstrate that an experienced NP can successfully and exponentially increase access to oncology specialist care.

*Consultative model:* In this model, the NP has a distinct role as a consultant. At CCI, this model involves pre-autologous transplant (auto-BMT) screening and workup. The majority of patients referred to auto-BMT have relapsed lymphoma and are heavily pre-treated with cardiotoxic chemotherapy and/or radiation to thorax with concurrent pre-existing comorbidities. At initial NP consultation, a comprehensive review is performed to identify comorbid conditions and cardiovascular risk factors and confirm their safety and eligibility for auto-BMT [15]. Complete history and physical exam are performed by the NP, along with additional screening and investigations to further elucidate risk profiles as indicated. The electronic medical record document with the full assessment and recommendations for auto-BMT suitability are summarized for the hematology team, available province-wide. Once the consultation is complete, the NP initiates the stem cell mobilizing protocol, with ongoing surveillance resuming with the referring hematologist.

A trainee project was undertaken to review the cardio-oncology needs of patients assessed in one year of the Consultative clinic (n = 73) [16]. Ethical approval was secured from the Health Research Ethics Board of Alberta Cancer, file HREBA.CC-19-0439. Following NP assessment and consultation, 16 (20%) required cardio-oncology interventions preventing eligibility for auto-BMT. This early identification allowed the cardio-oncology team to rapidly initiate pharmacotherapy and follow-up cardiac imaging to evaluate treatment effect [17]. As a result of these early assessments and interventions, 100% of the patients proceeded safely through auto-BMT [16]. This model has been established for well over a decade, and from 2014 to 2024, 854 auto-BMT consultations were performed by the NP, contributing substantially to the efficiency of the hematology team and overall auto-BMT safety.

*Partnered model:* In this setting, the NP and oncologist work ‘side by side’ assessing the next scheduled and unselected cancer patient. Similar skill sets allow the NP and MO to efficiently provide care to complex patients. ESAS and KPS are shown in Table 3. Should a patient issue arise that needs the immediate attention and time commitment of the MO (i.e., advancing symptom management or disease progression), the NP partner continues with the clinic and the remaining patients. Not only does this approach provide continuity, but it also decreases pressure on the MO allowing for important, sensitive discussions to occur without the stress of a clinic backing up.

This has been a successful model in a palliative radiation clinic where the NP and radiation oncologist (RO) performed similar activities with the exception of prescribing radiation. Radiation prescription is shown in Table 4. Ethical approval was secured from the Health Research Ethics Board of Alberta, Cancer, File HREBA.CC-21-0202. We reviewed patients attending this clinic from January 2008 to December 2010, with patients assessed by the first available provider. Symptom acuity was assessed by the Edmonton Symptom Assessment System (ESAS) and Karnofsky Performance Status (KPS), with higher scores indicating worse symptoms and functioning. Here we observed equivalent symptom severity among patients between both providers [18].

Moreover, no differences in the need for radiation were observed between groups, suggesting equivalent severity patterns of bone metastases. By working as a team, the NP further contributed by consulting on 29 patients who were ultimately ineligible for RT, allowing the RO to deploy their specialized skills more efficiently. Moreover, the work performed by the NP allowed 58% (137/235) more patients to access the clinic and receive consultation than if the RO was working alone (98/235) [18]. This model can be highly effective in large and/or complex patient populations where the NP and MO/RO have similar clinical and experiential skills.

*Most Responsible Provider (MRP) model:* This approach has been successfully implemented at CCI in the setting of chronic lymphocytic leukemia (CLL). The NP, as the MRP, assumes primary responsibility and ongoing consistent care during the patient’s course of treatment from diagnosis, treatment determination, prescribing, necessary diagnostics, and, when indicated, discharge from the program. Following the initial comprehensive consultation the surveillance plan, indicators for treatment initiation and self-help approaches are discussed and augmented at subsequent clinic visits.

Given that only ⅓ of patients with CLL low-risk disease will need treatment within the next 10 years, the NP focuses on the quality and quantity of future life, with the adage of ‘M&M: Monitoring and Moving On [19]’. If and when treatment is indicated per international criteria [20,21,22], the NP presents the case at tumor group rounds. Team treatment recommendations are documented in the chart for presentation to the patient. The NP orders initial treatment and subsequent follow-up. Hematology colleagues are available should complex concerns requiring discussion arise. The relevant NP has an annual case load of 400 CLL patients (2 half day clinics weekly) alongside other consultative and partnered model practices.

## 3. Discussion

The advancement of novel cancer treatments is resulting in a tsunami of patients and survivors requiring specialist cancer care far longer than ever before. These advancements have direct implications with not only increased wait times but for those within the system requiring continued oncology expertise far beyond what can be provided by existing MO and/or RO resources. To the best of our knowledge, we are the first to demonstrate how multiple NP models of care within one organization can substantially contribute to optimal patient cancer care and significantly alleviate ongoing access needs.

NPs were originally deployed in Canada to address physician shortages in the early 1970s in geographically challenging locations, where health service access was necessary [23,24]. Since then, the NP role has advanced substantially including geographical positioning, educational preparation, registration requirements, and scope of practice. Canadian regulations stipulate that all NPs possess a minimum of a Masters’ degree specific to NP practice, inclusive of extended supervised preceptorship [25]. NP-led care has been shown to be equivalent to physician care in multiple care settings with improved patient-reported outcomes and cost-effectiveness [26,27,28]. In a qualitative study, we found that despite traditional misunderstandings about health professional roles (MO and NP) cancer patients reported appreciating the benefits of NP care [29]. The models in our study highlight the potential for multiple future roles and serve as exemplars of the flexibility of NP care to meet the evolving needs of cancer patients, the multidisciplinary care team, and the health system.

Why was the CCI NP initiative particularly successful? We believe it was a timely intersection of visionary administration; observing the system demanding more than it was traditionally able to provide. Provincial funding cutbacks meant achieving more with less. Registered nurses with decades of oncology experience completed their NP clinical preceptorships with physicians at CCI, essentially training into their NP positions. These integrative, on-site preceptorships fostered acceptance and professional relationships with MO and RO physicians. The organization maximized the scope of practice of the entire care team, assigning the right provider with the right skill set to the right patient with specific needs that met that skill set. It was also understood that similar to MOs or RO practice, NPs must be assigned to specific tumor groups promoting the specialized knowledge associated with increasingly complex treatment modalities.

NP oncology care at the CCI is a compelling example of progress in a specialty field, serving as exemplars for other settings across Canada. Future work should focus on preparatory specialty fellowships similar to that of physicians and other health professionals. This is especially relevant, as the Canadian Council of Registered Nurse Regulators has mandated all NP graduate curricula focus on community-centered practice across life span, not specialty settings such as oncology. The NPs in the CCI clinics had decades of prior nursing experience in addition to the required master’s degree imparting significant disease-based knowledge at the outset. Similarly to medical residency, post-graduate training for newly graduated NPs will be required to meet the needs of increasingly complex oncology populations, a worthwhile investment.

Integration within health system leadership is essential for NP success. One solution is an NP representative at the leadership ‘table’ who can contribute to understanding the scope and contribution of NP practice and system-wide decision-making. Another could be an NP team lead, with a role as the site contact and representative for the NP group or the specific tumor group. At CCI, the leadership understood that NP responsibilities, similar to those of MO or RO, required administrative representation, a dedicated office space, administrative assistants, reserved education, and research time. In jurisdictions where NP care was not successfully adopted, issues included provider and organizational resistance to change, inappropriate or antiquated funding models, limited knowledge of the advanced nature of the role, and a lack of evidence-based health care advancement [26,27,30,31,32]. System-wide and multidisciplinary health professional training including administrators and physicians will promote better understanding of the potential contributions of NPs to oncology care in all settings. Taken together, oncology NPs hold significant potential to address the rapidly expanding care needs of a rapidly growing population of cancer patients and survivors today and in the future.

## Figures and Tables

**Table 1 curroncol-32-00492-t001:** Baseline characteristics of stage I and II female breast cancer patients 2012–2013 by active treatment models.

		Assigned		Available		Rural		Total	
		N	%	N	%	N	%	N	%
Patients 2012–2013		505	47.60	414	40.00	143	13.50	1062	
Characteristic									
Age	Median = 53 (Range: 24–78)								
	Less than 40	54	10.69	46	11.11	13	9.09	113	10.64
	40 to 49	147	29.11	126	30.43	40	27.97	313	29.47
	50 to 59	169	33.47	138	33.33	43	30.07	350	32.96
	60 to 69	126	24.95	90	21.74	39	27.27	255	24.01
	70+	9	1.78	14	3.38	8	5.59	31	2.92
Stage									
	I	193	38.22	129	31.16	35	24.48	357	33.62
	II	312	61.78	285	68.84	108	75.52	705	66.38
Morphology Grade									
	1	34	6.73	20	4.83	12	8.39	66	6.21
	2	159	31.49	120	28.99	56	39.16	335	31.54
	3	311	61.58	271	65.46	71	49.65	653	61.49
	4	1	0.20	0	0.00	0	0.00	1	0.09
	9	0	0.00	3	0.72	4	2.80	7	0.66
Lymphovascular Invasion									
	1-YES	214	42.38	172	41.55	48	33.57	434	40.87
	2-NO	284	56.24	235	56.76	84	58.74	603	56.78
	9-UNK	7	1.39	7	1.69	11	7.69	25	2.35
ER									
	0-Negative	111	21.98	86	20.77	32	22.38	229	21.56
	1-Positive	393	77.82	327	78.99	111	77.62	831	78.25
	9-Unknown	1	0.20	1	0.24	0	0.00	2	0.19
PR									
	0-Negative	172	34.06	128	30.92	51	35.66	351	33.05
	1-Positive	332	65.74	285	68.84	92	64.34	709	66.76
	9-Unknown	1	0.20	1	0.24	0	0.00	2	0.19
HER_2									
	0-Negative	415	82.18	305	73.67	116	81.12	836	78.72
	1-Positive	89	17.62	108	26.09	27	18.88	224	21.09
	9-Unknown	1	0.20	1	0.24	0	0.00	2	0.19
Comorbidity Group									
	10–19	184	36.44	139	33.57	47	32.87	370	34.84
	20–49	132	26.14	124	29.95	41	28.67	297	27.97
	50–69	179	35.45	148	35.75	54	37.76	381	35.88
	70–99	5	0.99	1	0.24	1	0.70	7	0.66
	NA	5	0.99	2	0.48	0	0.00	7	0.66
Total		505		414		143		1062	

**Table 2 curroncol-32-00492-t002:** Clinical and safety outcomes associated with early breast cancer active treatment models.

Model	Assigned		Available		Rural		Total		*p*-Value
	N	%	N	%	N	%	N	%	
Average Dose Intensity = total dose received/(max(dose) ∗ planned # of cycles)									<0.05
100%	159	31.49	102	24.64	41	28.67	302	28.44	
>90%	174	34.46	177	42.75	42	29.37	393	37.01	
<90%	172	34.06	135	32.61	60	41.96	367	34.56	
Treatment Completion = actual cycle/planned cycle									0.108
100%	380	75.25	305	73.67	95	66.43	780	73.45	
<100%	125	24.75	109	26.33	48	33.57	282	26.55	

**Table 3 curroncol-32-00492-t003:** ESAS and KPS, Mean, and Standard Deviation.

	NP (n = 137)	RO (n = 98)	*p* Value
Pain	6.1 (2.8)	5.3 (3)	0.067
Fatigue	5.4 (2.8)	5.7 (2.7)	0.391
Appetite	4.9 (3)	4.7 (3.2)	0.622
Wellbeing	4.2 (3)	4 (2.8)	0.471
Depression	2.8 (2.7)	2.8 (3)	0.953
Shortness of breath	2.7 (3)	2.7 (2.8)	0.892
Anxiety	2.6 (2.7)	3.2 (2.9)	0.118
Nausea	1.2 (2.1)	1.8 (2.5)	0.174
KPS	60.4 (17.3)	56.4 (24.4)	0.191

**Table 4 curroncol-32-00492-t004:** Radiation Prescription.

Provider	NP (n = 137)	RO (n = 98)	*p* Value
RT no	29	26	0.402
RT yes	108	72	0.643

## Data Availability

No new data were created as part of this paper.

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
