# Peer review of "Nurse Practitioner Care Delivery Models: Meeting the Rapidly Expanding Needs of Cancer Patients"

_curroncol, 2025, doi:10.3390/curroncol32090492_

Round 1
Reviewer 1 Report
Comments and Suggestions for Authors
The study aimed to describe four NP care models: to enhanced and expanded cancer care delivery. It is important with quick care processes in the cancer care, and a delay of even four weeks in starting cancer treatment (surgical, systemic, or radiotherapy) can increase survival rates. The initiated program to address gaps and involving nurse practitioners in cancer care is interesting and a good initiative. Some comments,
The introduction is short, and a consideration is whether this needs to be deepened further. And include the description of the model, as well as clarify the level of knowledge and competence that NP includes.
Would recommend that the method section is clarified, which design is current. Now the text about the models comes quite directly in the method, which would probably be good to introduce more clearly.
In the method it is written that data were collected from the Cancer Care EMR, with dose intensity and treatment completion calculated based on the projected versus actual dose and cycle administered for each chemotherapy agent. It is not a review type of study? Please clarify. If it is an empirical study, the method probably needs to contain a more structured order for headings, e.g. data collection, procedure, analysis.
Would it be valuable to show data/results as Table 1, from different models.
Consider printing abbreviations, for example in the Table so that it can be read independently.
Page 4, line 111-114; You write “To the best of our knowledge this is the first report of NP-led care in oncology demonstrated as safe and equivalent to that of MO providers specific to patient complexity….” Is this rationale and the knowledge gap that needs to be written into the background and clarified before the purpose?
Consider clarifying the effectiveness, in fostering acceptance and professional relationships between NPs and the doctors? Do you have any measurable outcomes or comments from participants as need to be included.
Do the malignancies impact the quality of care and outcomes for the patients, and do you have any results?
In focus of transferability in an international context, it would be valuable to highlight this more in the discussion. And also clarify identified barriers to the successful adoption of NP care in cancer care. Do you have examples where barriers have been overcome and the successful aspects.
Consider including a conclusion and also future directions and implementation.
The care needs of cancer survivors, and specific future strategies and activities that should be prioritized to enhance NP contributions to cancer care and oncology departments/settings.
Author Response
The study aimed to describe four NP care models: to enhanced and expanded cancer care delivery. It is important with quick care processes in the cancer care, and a delay of even four weeks in starting cancer treatment (surgical, systemic, or radiotherapy) can increase survival rates. The initiated program to address gaps and involving nurse practitioners in cancer care is interesting and a good initiative.
Some comments,
The introduction is short, and a consideration is whether this needs to be deepened further. And include the description of the model, as well as clarify the level of knowledge and competence that NP includes.
Would recommend that the method section is clarified, which design is current. Now the text about the models comes quite directly in the method, which would probably be good to introduce more clearly.
In the method it is written that data were collected from the Cancer Care EMR, with dose intensity and treatment completion calculated based on the projected versus actual dose and cycle administered for each chemotherapy agent. It is not a review type of study? Please clarify. If it is an empirical study, the method probably needs to contain a more structured order for headings, e.g. data collection, procedure, analysis. We agree, and have changed the manuscript type from “Review” to “Analysis”.
Would it be valuable to show data/results as Table 1, from different models. We agree, but given the differences in the Models we were unable to do so.
Consider printing abbreviations, for example in the Table so that it can be read independently.
Page 4, line 111-114; You write “To the best of our knowledge this is the first report of NP-led care in oncology demonstrated as safe and equivalent to that of MO providers specific to patient complexity….” Is this rationale and the knowledge gap that needs to be written into the background and clarified before the purpose?
Consider clarifying the effectiveness, in fostering acceptance and professional relationships between NPs and the doctors? Do you have any measurable outcomes or comments from participants as need to be included. We did not measure this question at the time of data collection.
Do the malignancies impact the quality of care and outcomes for the patients, and do you have any results? We do not know if different malignancies present differences in quality or outcomes at this time. The patient populations were considerably different (curative vs palliative, tumor types, systemic vs radiation treatment).
In focus of transferability in an international context, it would be valuable to highlight this more in the discussion. And also clarify identified barriers to the successful adoption of NP care in cancer care. Do you have examples where barriers have been overcome and the successful aspects. We did not collect data on the specific barriers/facilitators of each Model.
Consider including a conclusion and also future directions and implementation. The care needs of cancer survivors, and specific future strategies and activities that should be prioritized to enhance NP contributions to cancer care and oncology departments/settings. Done.
Reviewer 2 Report
Comments and Suggestions for Authors
Title: Oncology Nurse Practitioners: Meeting the Rapidly Expanding Needs of Cancer Patients
Summary: This manuscript, submitted to Current Oncology, evaluates the role of nurse practitioners (NPs) in addressing the increasing demand for oncology care in Alberta, Canada, particularly at the Cross Cancer Institute (CCI). It describes four NP care delivery models—Assigned, Consultative, Partner, and Most Responsible Provider (MRP)—implemented since 2004 to improve access to cancer care amidst a growing patient population and stagnant specialist numbers. The study highlights the effectiveness of these models in managing early breast cancer (EBC), autologous bone marrow transplant (auto-BMT) screening, palliative radiation, and chronic lymphocytic leukemia (CLL) care. A retrospective analysis of EBC patients (2012–2013) compared the Assigned model to Available and Rural models, focusing on chemotherapy dose intensity and treatment completion. The study also presents outcomes from the Consultative, Partner, and MRP models, emphasizing NPs' contributions to care efficiency and patient outcomes.
Importance: The study addresses a critical issue in oncology: the gap between rising cancer care demands and limited specialist resources. By demonstrating the efficacy of NP-led models, it offers a scalable solution for healthcare systems facing similar challenges, particularly in regions with recruitment difficulties for medical oncologists (MOs) and radiation oncologists (ROs).
Main Findings:
-
Assigned Model: In EBC, NP-led care achieved equivalent chemotherapy dose intensity and treatment completion compared to MO-led care, with no significant differences in patient complexity or outcomes.
-
Consultative Model: NP-led pre-auto-BMT screening identified 20% of patients needing cardio-oncology interventions, enabling safe transplant procedures for all assessed patients.
-
Partner Model: NPs working alongside ROs in palliative radiation clinics increased clinic capacity by 58%, with equivalent symptom acuity and radiation needs.
-
MRP Model: NPs managed 400 CLL patients annually, providing comprehensive care from diagnosis to discharge, with hematologist support for complex cases.
Main Conclusion: NP-led care models are safe, effective, and equivalent to physician-led care, significantly enhancing access to oncology services and addressing the growing needs of cancer patients and survivors.
Feedback and Comments to Authors
1. Importance, Originality, Title, and Abstract
-
Importance and Originality: The study is highly relevant to Current Oncology readers, including oncologists, NPs, and healthcare administrators, as it addresses a pressing global challenge: ensuring timely and effective cancer care amidst resource constraints. The originality lies in its detailed comparison of four distinct NP care models within one institution, a novel approach not widely reported in oncology literature. This provides a blueprint for other healthcare systems to adopt NP-led care.
-
Title: The title is clear and reflects the study’s focus on NPs addressing cancer care needs. However, it could be more specific by including “Care Delivery Models” to highlight the study’s core contribution (e.g., “Oncology Nurse Practitioner Care Delivery Models: Meeting the Rapidly Expanding Needs of Cancer Patients”).
-
Abstract: The abstract adequately summarizes the study’s purpose, methods, and key findings but could be improved by:
-
Specifying the retrospective nature of the EBC analysis and the descriptive approach for other models.
-
Quantifying key outcomes (e.g., “58% increased clinic capacity” or “100% safe auto-BMT procedures”).
-
Clarifying the study’s scope (e.g., single-center focus at CCI).
-
Including a stronger concluding statement on scalability or policy implications.
-
Recommendation: Revise the title for specificity and enhance the abstract with quantitative outcomes and methodological clarity.
2. Appropriateness of Study Approach and Experimental Design
-
Approach: The study combines a retrospective cohort analysis (for the Assigned model in EBC) with descriptive case studies of three other NP models (Consultative, Partner, MRP). This mixed approach is appropriate for evaluating both quantitative outcomes (e.g., dose intensity) and qualitative impacts (e.g., clinic efficiency).
-
Design:
-
The retrospective cohort design for EBC is suitable for comparing NP-led (Assigned) versus other models (Available, Rural) using registry data, given the high cure rates of EBC and clear outcome measures (dose intensity, treatment completion).
-
Descriptive analyses of the other models rely on historical data and case studies, which are appropriate for demonstrating feasibility and impact but limit generalizability.
-
-
Limitations: The study does not mention compliance with reporting guidelines (e.g., STROBE for observational studies). Confirming adherence to such standards would strengthen credibility. There is no mention of IRB approval or clinical trial registration, which is critical for retrospective studies using patient data.
Recommendation: Specify compliance with STROBE guidelines and clarify IRB approval status. Consider framing the descriptive components as a case series to align with study design terminology.
3. Appropriateness and Reproducibility of Data Collection and Techniques
-
Data Collection:
-
For the Assigned model, data from the Alberta Cancer Registry and Cancer Care EMR are well-described, with clear definitions of dose intensity and treatment completion. The use of Aggregated Clinical Risk Scores (ACRG) for comorbidity adjustment is robust and validated.
-
For other models, data collection is less detailed (e.g., Consultative model cites a trainee project with n=73; Partner model uses 2008–2010 data). Reproducibility is limited due to vague descriptions of data sources and collection methods.
-
-
Techniques: The retrospective analysis of EBC uses standard statistical methods (multivariate logistic regression, Hosmer-Lemeshow test), enhancing reproducibility. However, the descriptive models rely on narrative summaries, which are harder to replicate without standardized protocols.
-
Compliance: The study does not explicitly state compliance with CONSORT, PRISMA, or REMARK, though STROBE is relevant for the retrospective cohort. IRB approval or exemption status is not mentioned, which is a significant oversight for patient data studies.
Recommendation: Provide detailed data collection protocols for all models, confirm IRB approval, and align with STROBE guidelines for the retrospective component.
4. Analysis and Interpretation of Data
-
Analysis:
-
The EBC analysis uses multivariate logistic regression to assess dose intensity and treatment completion, adjusting for confounders (age, stage, ER/PR/HER2 status, lymphovascular invasion, ACRG). The use of SAS v9.1 and a p-value threshold of <0.05 is standard, though the choice of <0.5 in the text (likely a typo) should be corrected to <0.05.
-
Descriptive analyses for other models are less rigorous, relying on percentages and narrative outcomes (e.g., 58% increased clinic capacity, 100% safe auto-BMT). These lack statistical comparisons, limiting their strength.
-
-
Interpretation: The authors appropriately conclude that NP-led care is safe and equivalent to MO-led care based on EBC data. The descriptive models support claims of increased access and efficiency but are less conclusive due to the lack of controlled comparisons. The discussion of NP training and system integration is insightful but slightly speculative without supporting data.
-
Agreement with Conclusions: The conclusions are supported for the Assigned model due to robust statistical analysis. For other models, claims of effectiveness are plausible but require more rigorous evaluation to match the EBC analysis.
Recommendation: Correct the p-value typo, strengthen descriptive analyses with additional quantitative data (if available), and temper speculative claims in the discussion (e.g., on NP training needs).
5. Weaknesses and Limitations
-
Selection Bias: The retrospective EBC analysis may be subject to selection bias, as patients were not randomized to care models. The authors note similar clinical characteristics across models, but unmeasured confounders (e.g., patient preferences) could affect outcomes.
-
Sample Size: The EBC cohort (n=1062) is adequately powered, but the Consultative (n=73) and Partner (n=235) analyses use smaller samples, limiting statistical power.
-
Missing Data: The manuscript does not address missing data handling, which is critical for registry-based studies.
-
Single-Center Focus: The study is limited to CCI, reducing generalizability to other settings with different resources or NP training.
-
Lack of Patient-Reported Outcomes: The study focuses on clinical outcomes but lacks patient satisfaction or quality-of-life data, which could strengthen claims about NP care benefits.
Recommendation: Acknowledge these limitations in the discussion, particularly selection bias, missing data, and generalizability. Consider adding patient-reported outcome data if available.
6. Writing and Organization
-
Writing: The manuscript is generally clear but occasionally wordy, particularly in the discussion (e.g., “tsunami of patients” could be replaced with “surge”). The language is professional and suitable for an academic audience.
-
Organization: The structure is logical, with clear sections for each NP model. However, the transition between the retrospective EBC analysis and descriptive models feels abrupt. Subheadings within the Methods and Results sections could improve readability.
-
Clarity: The introduction effectively sets the context, but the methods section lacks detail for non-EBC models. The discussion could better integrate findings across all models to emphasize their collective impact.
Recommendation: Streamline wordy phrases, add subheadings for clarity, and provide more methodological detail for descriptive models.
7. Necessity and Clarity of Figures and Tables
-
Tables:
-
Table 1 (baseline characteristics) is clear, comprehensive, and stands alone, providing essential demographic and clinical data.
-
Table 2 (clinical outcomes) is well-organized, showing dose intensity and treatment completion across models with p-values. However, the p-value column could specify which comparisons are significant.
-
-
Necessity: Both tables are necessary to support the EBC analysis. No figures are included, which is appropriate given the data-driven focus.
-
Independence: Both tables are self-explanatory, with clear captions and data presentation, though adding a footnote to Table 2 clarifying statistical comparisons would enhance clarity.
Recommendation: Retain both tables, add a footnote to Table 2 for statistical clarity, and consider a figure (e.g., flowchart of NP models) to visually summarize the four care models.
8. Formal Statistical Review
-
EBC Analysis:
-
Strengths: The use of multivariate logistic regression adjusted for confounders (age, stage, ER/PR/HER2, ACRG) is robust. The Hosmer-Lemeshow test for goodness of fit is appropriate, though results are not reported. The sample size (n=1062) supports adequate power for detecting differences in dose intensity and treatment completion.
-
Issues:
-
The p-value threshold is listed as <0.5 (line 92), likely a typo for <0.05.
-
The manuscript does not report effect sizes (e.g., odds ratios) for logistic regression, which would strengthen interpretation.
-
Missing data handling is not described, which is critical for registry-based studies.
-
The Rural model’s lower dose intensity is attributed to provider variability, but no statistical test confirms this hypothesis.
-
-
-
Descriptive Models:
-
The lack of statistical analysis for Consultative, Partner, and MRP models weakens their evidential strength. Percentages (e.g., 58% increased capacity) are reported without confidence intervals or comparative statistics.
-
-
Recommendations:
-
Correct the p-value typo to <0.05.
-
Report odds ratios and confidence intervals for the EBC logistic regression.
-
Describe missing data handling (e.g., imputation or exclusion).
-
Add statistical comparisons for descriptive models, if feasible, or acknowledge their absence as a limitation.
-
9. Readability and English Writing Quality
-
The manuscript is readable, with clear sentences and appropriate academic tone. Minor issues include:
-
Wordy phrases (e.g., “far beyond what can be provided” could be “exceeding available resources”).
-
Inconsistent abbreviation use (e.g., “MO” and “RO” are defined late; “auto-BMT” is used before definition).
-
The typo “<0.5” for p-value (line 92) undermines credibility.
-
-
The English quality is high, with no major grammatical errors, but polishing for conciseness would improve flow.
Recommendation: Revise for conciseness, define abbreviations at first use, and proofread for typos (e.g., p-value).
10. Relevant Citations
Include the these citations in the discussion to support the Partner model and broaden the literature context considering various strategies in ameliorating side effects of oncologic treatments
A) Moezian GSA, et al. Oral silymarin formulation efficacy in management of AC-T protocol induced hepatotoxicity in breast cancer patients: A randomized, triple blind, placebo-controlled clinical trial. J Oncol Pharm Pract. 2022 Jun;28(4):827-835. doi: 10.1177/10781552211006182.
B) Ebrahimi N, et al. Randomized, Double-Blind, Placebo-Controlled Clinical Trial of Concurrent Use of Crocin During Chemoradiation for Esophageal Squamous Cell Carcinoma. Cancer Invest. 2024 Feb;42(2):155-164. doi: 10.1080/07357907.2024.2319754. C) Salek R, et al. Amelioration of anxiety, depression, and chemotherapy related toxicity after crocin administration during chemotherapy of breast cancer: A double blind, randomized clinical trial. Phytother Res. 2021 Sep;35(9):5143-5153. doi: 10.1002/ptr.7180. D) Sedighi Pashaki A, et al. A Randomized, Controlled, Parallel-Group, Trial on the Effects of Melatonin on Fatigue Associated with Breast Cancer and Its Adjuvant Treatments. Integr Cancer Ther. 2021 Jan-Dec;20:1534735420988343. doi: 10.1177/1534735420988343. E) Sedighi Pashaki A, et al. A Randomized, Controlled, Parallel-Group, Trial on the Long-term Effects of Melatonin on Fatigue Associated With Breast Cancer and Its Adjuvant Treatments. Integr Cancer Ther. 2023 Jan-Dec;22:15347354231168624. doi: 10.1177/15347354231168624.
11. Final Recommendation
The manuscript is a valuable contribution to oncology literature, demonstrating the efficacy and scalability of NP-led care models in addressing cancer care access challenges. The retrospective EBC analysis is robust, but the descriptive models need stronger methodological detail and statistical support. Minor revisions are needed to address typos, enhance the abstract, and clarify limitations.
Recommendation: Accept with Minor Revisions
-
Revise the title and abstract for specificity and quantitative emphasis.
-
Confirm IRB approval and STROBE compliance.
-
Correct statistical errors (e.g., p-value typo) and report effect sizes.
-
Provide detailed data collection methods for descriptive models.
-
Acknowledge limitations (e.g., selection bias, generalizability).
-
Streamline writing and add a figure to summarize NP models.
Author Response
Summary: This manuscript, submitted to Current Oncology, evaluates the role of nurse practitioners (NPs) in addressing the increasing demand for oncology care in Alberta, Canada, particularly at the Cross Cancer Institute (CCI). It describes four NP care delivery models—Assigned, Consultative, Partner, and Most Responsible Provider (MRP)—implemented since 2004 to improve access to cancer care amidst a growing patient population and stagnant specialist numbers. The study highlights the effectiveness of these models in managing early breast cancer (EBC), autologous bone marrow transplant (auto-BMT) screening, palliative radiation, and chronic lymphocytic leukemia (CLL) care. A retrospective analysis of EBC patients (2012–2013) compared the Assigned model to Available and Rural models, focusing on chemotherapy dose intensity and treatment completion. The study also presents outcomes from the Consultative, Partner, and MRP models, emphasizing NPs' contributions to care efficiency and patient outcomes. Importance: The study addresses a critical issue in oncology: the gap between rising cancer care demands and limited specialist resources. By demonstrating the efficacy of NP-led models, it offers a scalable solution for healthcare systems facing similar challenges, particularly in regions with recruitment difficulties for medical oncologists (MOs) and radiation oncologists (ROs).
Feedback and Comments to Authors
1. Importance, Originality, Title, and Abstract
Importance and Originality: The study is highly relevant to Current Oncology readers, including oncologists, NPs, and healthcare administrators, as it addresses a pressing global challenge: ensuring timely and effective cancer care amidst resource constraints. The originality lies in its detailed comparison of four distinct NP care models within one institution, a novel approach not widely reported in oncology literature. This provides a blueprint for other healthcare systems to adopt NP-led care.
Title: The title is clear and reflects the study’s focus on NPs addressing cancer care needs. However, it could be more specific by including “Care Delivery Models” to highlight the study’s core contribution (e.g., “Oncology Nurse Practitioner Care Delivery Models: Meeting the Rapidly Expanding Needs of Cancer Patients”). Agree, done.
Abstract: The abstract adequately summarizes the study’s purpose, methods, and key findings but could be improved by:
Specifying the retrospective nature of the EBC analysis and the descriptive approach for other models. Quantifying key outcomes (e.g., “58% increased clinic capacity” or “100% safe auto-BMT procedures”). Clarifying the study’s scope (e.g., single-center focus at CCI). Including a stronger concluding statement on scalability or policy implications.
Recommendation: Revise the title for specificity and enhance the abstract with quantitative outcomes and methodological clarity. Agree, done.
2. Appropriateness of Study Approach and Experimental Design
Approach: The study combines a retrospective cohort analysis (for the Assigned model in EBC) with descriptive case studies of three other NP models (Consultative, Partner, MRP). This mixed approach is appropriate for evaluating both quantitative outcomes (e.g., dose intensity) and qualitative impacts (e.g., clinic efficiency).
Design:The retrospective cohort design for EBC is suitable for comparing NP-led (Assigned) versus other models (Available, Rural) using registry data, given the high cure rates of EBC and clear outcome measures (dose intensity, treatment completion).
Descriptive analyses of the other models rely on historical data and case studies, which are appropriate for demonstrating feasibility and impact but limit generalizability.
Limitations: The study does not mention compliance with reporting guidelines (e.g., STROBE for observational studies). Confirming adherence to such standards would strengthen credibility. There is no mention of IRB approval or clinical trial registration, which is critical for retrospective studies using patient data.
Recommendation: Specify compliance with STROBE guidelines and clarify IRB approval status. Consider framing the descriptive components as a case series to align with study design terminology.
Agree, done. We have clarified the manuscript type as ‘Article’ vs “Review’ for clarity for the reader.
3. Appropriateness and producibility of Data Collection and Techniques
Data Collection:For the Assigned model, data from the Alberta Cancer Registry and Cancer Care EMR are well-described, with clear definitions of dose intensity and treatment completion. The use of Aggregated Clinical Risk Scores (ACRG) for comorbidity adjustment is robust and validated.
For other models, data collection is less detailed (e.g., Consultative model cites a trainee project with n=73; Partner model uses 2008–2010 data). Reproducibility is limited due to vague descriptions of data sources and collection methods.
Techniques: The retrospective analysis of EBC uses standard statistical methods (multivariate logistic regression, Hosmer-Lemeshow test), enhancing reproducibility. However, the descriptive models rely on narrative summaries, which are harder to replicate without standardized protocols.
Compliance: The study does not explicitly state compliance with CONSORT, PRISMA, or REMARK, though STROBE is relevant for the retrospective cohort. IRB approval or exemption status is not mentioned, which is a significant oversight for patient data studies.
Recommendation: Provide detailed data collection protocols for all models, confirm IRB approval, and align with STROBE guidelines for the retrospective component. Agree, done.
4. Analysis and Interpretation of Data
Analysis:The EBC analysis uses multivariate logistic regression to assess dose intensity and treatment completion, adjusting for confounders (age, stage, ER/PR/HER2 status, lymphovascular invasion, ACRG). The use of SAS v9.1 and a p-value threshold of <0.05 is standard, though the choice of <0.5 in the text (likely a typo) should be corrected to <0.05.
Descriptive analyses for other models are less rigorous, relying on percentages and narrative outcomes (e.g., 58% increased clinic capacity, 100% safe auto-BMT). These lack statistical comparisons, limiting their strength.Interpretation: The authors appropriately conclude that NP-led care is safe and equivalent to MO-led care based on EBC data. The descriptive models support claims of increased access and efficiency but are less conclusive due to the lack of controlled comparisons. The discussion of NP training and system integration is insightful but slightly speculative without supporting data.
Agreement with Conclusions: The conclusions are supported for the Assigned model due to robust statistical analysis. For other models, claims of effectiveness are plausible but require more rigorous evaluation to match the EBC analysis.
Recommendation: Correct the p-value typo, strengthen descriptive analyses with additional quantitative data (if available), and temper speculative claims in the discussion (e.g., on NP training needs).
Correction completed, addition of odds ratios calculated from logistic regression. Regarding speculative claims, we specifically focused on NP training needs, as Canadian NP training is undergoing a dramatic change, with legislation requiring primary care training only. Specialty care training such as oncology will have to be planned after Masters degree completion, ideally structured as a fellowship or certification.
5. Weaknesses and Limitations
Selection Bias: The retrospective EBC analysis may be subject to selection bias, as patients were not randomized to care models. The authors note similar clinical characteristics across models, but unmeasured confounders (e.g., patient preferences) could affect outcomes. Missing Data: The manuscript does not address missing data handling, which is critical for registry-based studies. Given the variability of available care providers, randomization to a particular care provider was not possible. Perhaps in the future this will be possible. To the best of our knowledge, all data for patients treated in the province of Alberta were available in the ARIA system. The patients included were stage I & II breast cancer patients who were diagnosed in 2012 and 2013, had no other primary within 5 years, had surgery and received chemotherapy within 85 days post surgery at Cancer Care Alberta.
Sample Size: The EBC cohort (n=1062) is adequately powered, but the Consultative (n=73) and Partner (n=235) analyses use smaller samples, limiting statistical power. We agree this is a limitation.
Single-Center Focus The study is limited to CCI, reducing generalizability to other settings with different resources or NP training. Yes, we agree this is a limitation.
Lack of Patient-Reported Outcomes: The study focuses on clinical outcomes but lacks patient satisfaction or quality-of-life data, which could strengthen claims about NP care benefits. Unfortunately PROMs were phased into ARIA well after these studies were performed, we agree that this information may have enriched our observations.
Recommendation: Acknowledge these limitations in the discussion, particularly selection bias, missing data, and generalizability. Consider adding patient-reported outcome data if available. Done, see above.
6. Writing and Organization
Writing: The manuscript is generally clear but occasionally wordy, particularly in the discussion (e.g., “tsunami of patients” could be replaced with “surge”). The language is professional and suitable for an academic audience. I was unable to find a word that adequately substituted ‘tsunami’, which is indeed the situation.
Organization: The structure is logical, with clear sections for each NP model. However, the transition between the retrospective EBC analysis and descriptive models feels abrupt. Subheadings within the Methods and Results sections could improve readability.
Clarity: The introduction effectively sets the context, but the methods section lacks detail for non-EBC models. The discussion could better integrate findings across all models to emphasize their collective impact.
Recommendation: Streamline wordy phrases, add subheadings for clarity, and provide more methodological detail for descriptive models. Agree, done.
7. Necessity and Clarity of Figures and Tables
Tables: Table 1 (baseline characteristics) is clear, comprehensive, and stands alone, providing essential demographic and clinical data. Table 2 (clinical outcomes) is well-organized, showing dose intensity and treatment completion across models with p-values. However, the p-value column could specify which comparisons are significant.
Necessity: Both tables are necessary to support the EBC analysis. No figures are included, which is appropriate given the data-driven focus.Independence: Both tables are self-explanatory, with clear captions and data presentation, though adding a footnote to Table 2 clarifying statistical comparisons would enhance clarity.
Recommendation: Retain both tables, add a footnote to Table 2 for statistical clarity, and consider a figure (e.g., flowchart of NP models) to visually summarize the four care models.
Tables remain, edited.
We are unable to develop a figure/flowchart given the wide differences in patient populations (curative vs palliative, tumor types, systemic vs radiation treatment).
8. Formal Statistical Review
EBC Analysis:
Strengths: The use of multivariate logistic regression adjusted for confounders (age, stage, ER/PR/HER2, ACRG) is robust. The Hosmer-Lemeshow test for goodness of fit is appropriate, though results are not reported. The sample size (n=1062) supports adequate power for detecting differences in dose intensity and treatment completion.
Issues:
The p-value threshold is listed as <0.5 (line 92), likely a typo for <0.05.The manuscript does not report effect sizes (e.g., odds ratios) for logistic regression, which would strengthen interpretation.Missing data handling is not described, which is critical for registry-based studies. See #5 above.
The Rural model’s lower dose intensity is attributed to provider variability, but no statistical test confirms this hypothesis. We agree with this limitation.
Descriptive Models: The lack of statistical analysis for Consultative, Partner, and MRP models weakens their evidential strength. Percentages (e.g., 58% increased capacity) are reported without confidence intervals or comparative statistics.
Recommendations:
Correct the p-value typo to <0.05.Report odds ratios and confidence intervals for the EBC logistic regression. Describe missing data handling (e.g., imputation or exclusion).Add statistical comparisons for descriptive models, if feasible, or acknowledge their absence as a limitation. Agree with comments, corrections completed. The descriptive models emulate real-world clinical care and were collected as such, therefore no additional statistics were performed.
9. Readability and English Writing Quality
The manuscript is readable, with clear sentences and appropriate academic tone. Minor issues include:
Wordy phrases (e.g., “far beyond what can be provided” could be “exceeding available resources”).
Inconsistent abbreviation use (e.g., “MO” and “RO” are defined late; “auto-BMT” is used before definition).
The typo “<0.5” for p-value (line 92) undermines credibility.
The English quality is high, with no major grammatical errors, but polishing for conciseness would improve flow.
Recommendation: Revise for conciseness, define abbreviations at first use, and proofread for typos (e.g., p-value).
Completed, see above.
10. Relevant Citations
Include the these citations in the discussion to support the Partner model and broaden the literature context considering various strategies in ameliorating side effects of oncologic treatments
A) Moezian GSA, et al. Oral silymarin formulation efficacy in management of AC-T protocol induced hepatotoxicity in breast cancer patients: A randomized, triple blind, placebo-controlled clinical trial. J Oncol Pharm Pract. 2022 Jun;28(4):827-835. doi: 10.1177/10781552211006182.B) Ebrahimi N, et al. Randomized, Double-Blind, Placebo-Controlled Clinical Trial of Concurrent Use of Crocin During Chemoradiation for Esophageal Squamous Cell Carcinoma. Cancer Invest. 2024 Feb;42(2):155-164. doi: 10.1080/07357907.2024.2319754. C) Salek R, et al. Amelioration of anxiety, depression, and chemotherapy related toxicity after crocin administration during chemotherapy of breast cancer: A double blind, randomized clinical trial. Phytother Res. 2021 Sep;35(9):5143-5153. doi: 10.1002/ptr.7180. D) Sedighi Pashaki A, et al. A Randomized, Controlled, Parallel-Group, Trial on the Effects of Melatonin on Fatigue Associated with Breast Cancer and Its Adjuvant Treatments. Integr Cancer Ther. 2021 Jan-Dec;20:1534735420988343. doi: 10.1177/1534735420988343. E) Sedighi Pashaki A, et al. A Randomized, Controlled, Parallel-Group, Trial on the Long-term Effects of Melatonin on Fatigue Associated With Breast Cancer and Its Adjuvant Treatments. Integr Cancer Ther. 2023 Jan-Dec;22:15347354231168624. doi: 10.1177/15347354231168624.
We are satisfied with the references as written. Our team is composed of national and international leaders in nurse practitioner care.
11. Final Recommendation
The manuscript is a valuable contribution to oncology literature, demonstrating the efficacy and scalability of NP-led care models in addressing cancer care access challenges. The retrospective EBC analysis is robust, but the descriptive models need stronger methodological detail and statistical support. Minor revisions are needed to address typos, enhance the abstract, and clarify limitations.
Recommendation: Accept with Minor Revisions
Revise the title and abstract for specificity and quantitative emphasis.
Confirm IRB approval and STROBE compliance.
Correct statistical errors (e.g., p-value typo) and report effect sizes.
Provide detailed data collection methods for descriptive models.
Acknowledge limitations (e.g., selection bias, generalizability).
Streamline writing and add a figure to summarize NP models.
Round 2
Reviewer 1 Report
Comments and Suggestions for Authors
Dear Authors,
Comments and revision have been addressed.